# Investigating Patterns of Research Collaboration and Citations in Science and Technology: A Case of Chiang Mai University

**Boontarika Paphawasit** [1,2,*] and **Ratapol Wudhikarn** [2,3]

1 Department of Modern Management and Information Technology, College of Arts, Media and Technology, Chiang Mai University, Chiang Mai 50200, Thailand

2 A Research Group of Modern Management and Information Technology, College of Arts, Media and Technology, Chiang Mai University, Chiang Mai 50200, Thailand; ratapol.w@cmu.ac.th

3 Department of Knowledge and Innovation Management, College of Arts, Media and Technology, Chiang Mai University, Chiang Mai 50200, Thailand

* Correspondence: boontarika.p@cmu.ac.th

**Abstract:** This study investigates 3883 articles published by researchers affiliated with Chiang Mai University in science and technology from January 2010 to December 2019 to test whether research team characteristics and collaboration patterns can determine a citation rate. Citations were retrieved from the Scopus database and compared with their (1) number of authors, (2) type of publication, (3) gender of authors, (4) SJR values, (5) country of international collaborators, (6) number of affiliated institutions, and (7) international diversity index. The findings were based on quantile regressions and indicated that the number of authors strongly influenced citations, which increases the likelihood of being cited. The citation advantage of being a foreign-first author only existed at the 0.25th quantile; however, the evidence of foreign-first author citation advantages or disadvantages for the moderate and very productive publications was not found. A significantly positive effect of SJR value on citations was found while being a female first author negatively impacted the citation rate. These findings can be used in the planning and managing process of producing scientific and technological research to improve the research quality, boost the research impact, and increase opportunities for research results to be utilized.

**Keywords:** research collaboration; team characteristics; diversity; gender; citations

## 1. Introduction

Scientific research aims at broadening the circle of knowledge in the light of the proficiency's acceleration and the information's proliferation; it is a measure of the advancement of society and nation, as the nation's wealth is not measured by natural wealth, but the productivity of human capital is serving the community. Research-related activities are activities that a definite process uses both intellectual and physical resources to accomplish since it is an academic activity that results from problem-solving and creativity. The systematic intellectual resources and the resulting outcomes benefit academic discovery, community development, social change, and industry advancement (Imhonopi and Urim 2013). Some countries prioritize various kinds of research because it has a vital role in developing human societies of different aspects of civilizational advancement. Today's problems, with a wide range of global impacts (e.g., green energy exploration, resource destruction, biodiversity damage), require advanced science and technology to deal with such problems. Science and technology are concerned with applying knowledge of nature to benefit human life and industrial development. Research in science and technology can help solve poverty, especially in underdeveloped countries, by producing products that meet basic human needs and serve as indicators of economic progress and political power. The crystallization of science and technology in nations has brought about economic, social, and political advancement, which significantly benefits the country's development. Science

and technology, research, and innovation are interrelated; research and innovation are robust if science is robust. Strengthening the infrastructure that supports research and technology services to endow the creation of scientific and technological knowledge is a matter for the authorities in each country. Several educational institutions and policymakers worldwide are focusing on accelerating academic circles in science and technology capabilities to meet current and future demands; therefore, it has become a policy of concern for the Department of Education worldwide (Lee et al. 2019). STEM-related research trends emerged and developed that have increased rapidly, with STEM publications in multiple journals rising between 2013 and 2017 (Lin et al. 2019), which is in line with the trend over the past three years of STEM research in Southeast Asia (Ha et al. 2020) and Europe and America over the past year (Li et al. 2020).

Since the research is a systematic and controlled study with a planned research design, the results must be practical and reliable; the essential qualification of a good researcher relates to the individual perception of the underlying research ideology. It is necessary to have an apparent involvement in research activities to be listed in the published article. Researchers must determine the research subjects and plan and design operational guidelines for conducting research. Writing a research project proposal is required by various funding sources to submit a request for financial support, and a cross-disciplinary project is one of the criteria to be considered. It was found that doing research as a small project has a weakness; it can be used for limited benefits because it lacks connection with other projects that can be used comprehensively. Researchers often work on topics in a particular field that is not fully integrated, so the research results are just a fraction of the problem, which is not practical. Therefore, research collaboration (RC) is essential because co-research encourages the area for coordinating ideas and physical activities to attain valuable conclusions. The concept behind research collaboration is derived from the proverb "two heads are better than one", in which more researchers working together have a better likelihood of solving a specific problem or creating new scientific knowledge (Katz and Martin 1997). Research Collaboration includes research networks, joint research, and the center of excellence, and it is commonly recognized as cross-institutional, cross-disciplinary, cross-country research collaboration. Such coordination enables seniority diversity, which is paramount to young researchers, as scientific research production demands diverse skills and proficiencies. Skills development is often generated through collaboration over some time. Therefore, such cooperation is acknowledged as team management contributing to skill development and transfer (Shaikh 2015).

The nature of the integrated team expands the dimensions of the research project. For example, it is research to solve the country's fundamental problems or lead to industrial production in which several individual projects are related or integrated and interdisciplinary. Such a project is carried out in an integrated and holistic manner and coordinated by the public and/or private sectors. The government's policy approach towards research should take the results of the country's administration and development according to the government's policy, including the current state of the country's research system, into consideration in formulating the country's research strategy to enable researchers to conduct integrated research effectively. Research collaboration is also indicative of the core competency and productivity of the university since it aims to disseminate new findings, exchange innovative learning, build research communities, and develop science across disciplines. Increasing research capacity is now crucial in developing educational institutions globally to become world-class universities. Institutions worldwide face international challenges to increase research productivity (e.g., producing quality research, creating new knowledge, publishing in high-quality journals, and being cited) and attempt to be ranked in the well-known university list; they continually develop and publish research, considering the research quality assessment an indicator of the institution's quality. Research output is essential for applying assessment for entry into higher academic positions for the individual researcher. In becoming a world-class university, research productivity is one of the university's missions. A multidisciplinary research environment positively affects

research creativeness and wisdom creation (Gibbons et al. 1994; Schmickl and Kieser 2008). Thus, funding bodies often encourage researchers to participate in university–industry collaboration, centers of excellence, multidisciplinary research institutes and centers, and industrial interdisciplinary research to create innovative knowledge (Bozeman and Corley 2004; Bozeman and Boardman 2014; Cummings and Kiesler 2005; Corley et al. 2006). This policy explains the growing number of research collaborations due to the increasing researcher's number finding and applying for grant funding (O'Brien 2012). Nowadays, Thai universities are constantly evolving to become world-class universities; therefore, this research strategy is vital in achieving this goal to be consistent with the country's policy of upgrading education and research. From the current university rankings, research productivity is one of the most critical factors in evaluating which universities stand out because research is the product of knowledge, which is distinctly abstract. According to information from the International Institute of Management Development (IMD), the urgent issues that should be addressed for Thailand's research are increasing investment in research and development, increasing the research and development personnel's quantity and quality, and encouraging the private sector to participate more in research and development (Open Development Thailand 2017). Thailand established a research management policy for creating research networks inside and outside the university, such as the Center of Excellence, Thailand's Research University Network. Many fora are organized for researchers from different faculties to coordinate integrated research, particularly the government and private sector funding. As a result, the establishment of the Center for Academic Excellence has created collaboration and built research networks across disciplines, faculties, and universities, which has resulted in networking with researchers at both national and international levels (Johnston et al. 2020).

Academics are interested in the results of collaborative research on research productivity, with indicators being the publication number and the citation rates. The study of indicators in this manner, a bibliometric analysis, employs published data to analyze statistical data to measure research quality and the potential of researchers and institutions (Agarwal et al. 2016). Chiang Mai University is an autonomous higher education institution in Thailand that holds a clear policy to strive for research development to gain higher competency for becoming a world-class university with the research promotion policies to create a reputation and academic excellence in various fields. One of the key policies is to promote the creation of facilities and resources and to create an academic environment and atmosphere to support quality research in all disciplines. The standard quantitative and qualitative measures of such a research strategy are research funding, the number of articles published in international journals such as the Web of Science (ISI) or the Scopus database, and regards citation counting, including the H-index. Citations are those that link current research with research that has been previously researched and can be traced back. In general, high-quality research has been cited instantaneously and received a high citation rate (Bornmann and Daniel 2010; Van Dalen and Henkens 2005). The number of citations is essential for assessing a research's quality (Kaplan et al. 2014) and directly affecting university rankings. Many factors influence the citation rates apart from the research contribution that discovers new knowledge with a proper research methodology: choosing to publish in journals of high quality and with a high reputation or high-impact factor are of concern. The nature of the individual research discipline, topic interest, open access, and writing style also influence the citation numbers because these factors might affect the accessibility of the research; it also affects the researchers' social relationships (Bornmann et al. 2008; Didegah and Thelwall 2013).

Research collaboration was previously the result of researchers' autonomous behavior; without outside contributors' intervention, researchers decide when and with whom to collaborate. Important goals such as increasing research capacity, building the country's competitiveness, dealing with challenging problems in the global society, and maintaining good international relations have become crucial for research cooperation (Boekholt et al. 2009). In this case, this study aimed to examine what characteristics of research teams

(i.e., first author, gender, number of team members, number of institutions participating in research, cooperation with foreign researchers, type of work published, and the international cooperation) influence research published in science and technology, and how they affect the research productivity; it focused on implementing research policies in team-level cooperation of the research published by professors and researchers in the science and technology field of Chiang Mai University. The data were collected from the Scopus database, and the quantile regression analysis was employed to study the impact of research collaboration on the citations of Chiang Mai University's researchers. The answer of whether collaborative research can produce higher quality research reflects the importance of research quality, in addition to answering scholastic questions; it creates value for academics or researchers within the team in monetary and non-monetary areas such as compensation, salary increases, awards, academic standing, and perceptions from people in academic circles. Therefore, it is an interesting topic for researchers of all fields, particularly scholars who do not have academic titles and junior researchers, as a guideline for creating research to be more productive. Based on the belief that cooperation is positively correlated with research quality, universities and funding institutions encourage collaborative research as a team and drive policy issues by offering incentive structures for joint publication. Thus, the findings of this study could provide meaningful answers to universities and funding institutions in support of researchers' collaborative research policies and can be further studied by expanding the scope or approach to increase diversity and depth of study regarding research management both nationally and internationally.

## 2. Literature Review

There has been a continuous increase in the size and emphasis of scientific cooperation in knowledge production in the 20th century. Sharing ideas and verifying scientific research findings with other scientists is imperative; scientific knowledge production has always been more sociable than in isolated institutions (Finholt and Olson 1997), and unprecedented levels of research collaboration stem from efforts to solve today's global problems, such as social, economic, technological transformations (Bozeman and Boardman 2014). Diverse types of indexes are applied to define successful research. At present, the indicators used for journal quality assessments include Impact Factor values from the Web of Science database, SCimago Journal Rank (SJR), Impact Per Paper (IPP), and Normalized Impact Per Paper (SNIP) values from the Scopus database. At the same time, assessing the quality of publishing journals also uses metrics built on counting the total citations for that journal over the past few years (Bornmann et al. 2012). The number of citations is calculated when the published research is re-cited by research that resides in a trusted database. Research on citation analysis has been increasingly studied, with content on citation analysis of journals published in various fields regarding direction and trends from past to present. Factors affecting citation values of published research were studied using bibliometrics (Agarwal et al. 2016). Recent research indicates that interesting and high-quality research is often citation-increasing (Bornmann and Daniel 2010) and published in journals with high SJR values or high quartiles. Despite the number of citations that come from the quality of research methodology and its contribution, it also depends on discipline, nature of research writing, researchers and research team characteristics, and journal selection (Bornmann et al. 2008).

According to the characteristics of the research team, various factors affect the research productivity in both quantitative and qualitative views (Abramo et al. 2009). Since research collaboration creates advantages in several forms; it stimulates intellectual benefits from information and knowledge exchange; it can lead to financial savings concerning training costs, physical facilities, and pertinent resources. Collaboration, in theory, is more effective due to the diverse skills of individuals in teams (Gibbons et al. 1994), since each individual has unique talents and abilities. When they come to work together and share for a common purpose, it can create a real competitive advantage (Murphy 2011). The research team did not happen randomly; the selection of participants for the research team is determined

by the ability of the researchers that the team expects. In this case, team characteristics (e.g., team size, international collaborative research, seniority diversity, and institutional diversity) play an essential role in building a research team to get the most influential group. Many researchers endeavor to pinpoint the association between co-authorship and its effect on research productivity.

It was found that co-authored research has a greater impact on productivity than single author publications regarding the publication number (Wuchty et al. 2007; Lee and Bozeman 2005; Sooryamoorthy 2017; Gazni and Didegah 2011). Articles with more authors have a propensity for attracting more citations, particularly in science (Frenken et al. 2005; Glänzel 2002; Leimu and Koricheva 2005; Persson et al. 2004; Wuchty et al. 2007). Co-authored papers with researchers within international collaborative teams were more likely to attract citations than national or regional collaborative research (Narin et al. 1991; Frenken et al. 2005). The international cooperation's citation impact is greater than the national one (Frenken et al. 2010), and international collaborators often receive large amounts of funding (Bozeman and Corley 2004). Durden and Perri (1995) utilized annual economics publications over 24 years as time-series data. The results revealed a positive relationship between the number of co-authored papers and the publication numbers; they also pointed out that cooperation boosts total publication productivity and article production per capita. An Italian multidisciplinary study pointed out that multiple-authored publications attracted more citations and indicated that collaboration was beneficial to research (Franceschet and Costantini 2010). A more extensive research team may be more inclined to create an article with more citations. Compared with other factors being constant, they were more likely to have more reputable or older scholars who had a greater impact on citations (Haslam et al. 2008)—the research team size may be less influential than the additional prominent researchers.

Despite the evidence supporting the positive effects of collaboration, a negative impact was found: increased travel costs, more transactions, and communication among team members. Nobody can guarantee that members will get the most from the exchange of facilities, and the success of a research project cannot be assured (Didegah and Thelwall 2013). Although decision-making quality improves with diverse team members brainstorming, large teams with heterogeneous members can make it difficult and time-consuming to reach a consensus when making decisions. As a result, interpersonal conflict increases with team size, thus impeding teamwork (Amason and Schweiger 1994). Diversity in the seniority and nationality of team members harms team outcomes (Gazni and Didegah 2011; Stvilia et al. 2010). The international collaboration had a greater influence to many extents. In contrast, inter-institutional cooperation did not impact publication productivity (Didegah and Thelwall 2013). In choosing collaborators, most researchers are not particularly universal as they manage to coordinate with individuals in their workgroups (Bozeman and Corley 2004).

Apart from the team's characteristics, it is typically comprehended that selecting the journal to be published is an essential factor that positively influences research citation values. Researchers who expect the published paper to be quickly and broadly referenced will need to publish in a peer-reviewed journal with a high quartile or a journal with a high Impact Factor or SJR. The literature results also pointed out that the citations were statistically significantly related to the journal quality indicators (Cartes-Velásquez and Manterola 2017; Didegah and Thelwall 2013); however, selecting a journal with a high reputation and high publishing demand reduces the likelihood of publishing success in cases where the research paper is of inferior quality. At the same time, publishing in a low-impact factor or SJR journal decreases the likelihood of research being cited or having fewer citations.

In this case, managing a team of authors plays a vital role in such collaborative situations. When multiple researchers work together on the same team, they need to share resources and support each other to get the job done. Notably, in research teams funded by government or private entities, the completion of projects on time affects subsequent

funding considerations; it challenges research team leaders in managing large, complicated collaborations to maximize the capabilities of team members together with mitigating the weaknesses caused by their constituents. Therefore, it is necessary to understand the characteristics of collaborative teams that may affect research efficiency; this comprehension allows team managers to manage their members to increase the quality of subsequent publications.

### 2.1. Number of Authors per Article

An issue that was investigated regarding the impact on research findings was the size of the team, as more research contributors are expected to complete the research project faster. Having more team members encourages more rigorous internal audits to correct errors, and the sharing of specialized knowledge and skills within the team increases the quality of research output. Therefore, multi-author publications are considered higher quality than solo papers. Literature backs this assumption because it has been found that the relationship between team size and scientific outcomes is positive (Beaver 2004; Lee and Bozeman 2005; Wuchty et al. 2007; Martín-Sempere et al. 2008; Sooryamoorthy 2009; Fischbach et al. 2011; Gazni and Didegah 2011; Fox et al. 2016). There is also substantial evidence that the number of published citations has increased as a result of the larger author team (Lawani 1986; Katz and Hicks 1997; Baldi 1998; Bornmann and Daniel 2010; Gazni and Didegah 2011; Annalingam et al. 2014; Biscaro and Giupponi 2014). A positive effect of the author numbers on the quality, length, and frequency of publications was found, and the correlation between co-authorship and individual outcomes was negative after the team downsized (Hollis 2001); however, there was also a negative correlation between research size and efficiency: having a larger team decreases the productivity of the team (Carayol and Matt 2006), and the bigger the team size, the higher the level of collaboration difficulty (Beaver 2004). Teams made up of multiple authors did not result in more citations (Medoff 2003; Hinnant et al. 2012; Bergh and Perry 2006). Consistent with Abramo et al. (2009)'s study, the strongly positive relationship between team sizes and research outcomes was uncovered only in industrial and information engineering; moreover, past research also concluded that a relationship between the number of authors and research productivity did not exist (Seglen and Aksnes 2000). Regarding team size, the hypothesis is indicated as follows.

**H1.** *Citation counts increase with the number of listed authors for a publication.*

### 2.2. Type of Research Article

A research article is a written work that takes information from a research report, compiled or summarized into a body of knowledge in a form presented to the reader to understand research problems and methods of research production, the findings, and the implications. Research papers may be presented at an academic conference in either an oral or poster format for publishing in academic conference proceedings—other types of the research article include papers published in academic journals. Publishing research papers must pass a review, and the contents are screened for being in the publishing criteria of particular journal standards by the assessment committee. The sources for disseminating research results are national academic conferences, international academic conferences, national academic journals, and international academic journals. The benefits of publishing research results will be for both researchers and the journal/proceeding itself; it is mutually advantageous and satisfactory to all parties involved in helping develop knowledge, since the research contributions will be disseminated and extended by other scholars for further study. The completed research project, particularly the project funded by the third parties, is required to be published in an accredited journal certifying that highly qualified experts have reviewed the research project according to the standards.

In other words, the project will not be completed if it is not accepted for publication in a suitable academic journal; it leads to the question of why it is necessary to publish in a quality academic journal instead of print media or conference proceedings. The reason

is that every article published in a quality academic journal is reviewed for scientific credibility and validity through an intensive review process by experts in the field and based on scientific principles, reliable research results, useful both in theory and in practice. The higher the quality of a journal, i.e., the higher the impact factor, the more difficult it is to be accepted for publication in such a journal. Once published, it demonstrates expertise in the field and the ability to conduct rational research, scientifically accurate and reliable processes of the research team, and the researchers' self-worth, which means reputation and journal quality reflect the researcher's standing.

Typically, the number of articles a researcher publishes in a reputable or high-quality journal is considered when necessary to decide a researcher's recruitment, performance assessment, advancement, or grant (Cunill et al. 2019). On the other hand, academics are also rigorously convinced of the quality of accredited research; therefore, publishing a paper in a journal, especially at the top quartile or in a reputable database, will attract more citations, resulting in a higher citation rate. For this reason, scholars try to manage their time resources to publish their research in academic journals. Regarding the analysis of citation patterns in various scientific fields, citations from conference proceeding is still a metric that many bibliographic researchers include in their studies. Despite the impact of journal articles being significantly higher than that of conference papers, the importance of this type of article could still be measured by the number of citations received (Michels et al. 2013).

Contrary to the assumption that journal articles receive more citations than conference papers, literature also reports that it depends on the discipline and the research topic, where the hot topics or impact issues are cited frequently. Published proceedings offer a chance to contend on a trending topic before others and depict the newest findings (González-Albo and Bordons 2011; Lin et al. 2014) since the journal's reviewing process takes a long time than conference papers, this offers better quality. For rapidly evolving fields such as computing and information science, presenting research results at symposiums or academic conferences is one of the key ways to disseminate information and knowledge as an additional source of standard journals (Glänzel et al. 2006; Shamir 2010; Makvandi et al. 2021). Although proceedings gain citations rapidly, they have a limited scientific impact and are outdated faster than journal articles (Lisée et al. 2008). According to Rahm (2008), conference papers receive considerable citing than journal articles in some areas of Computer Science, such as the database. Apart from directly submitting the paper to the desired journal, extending conference work for journal publication is another path to carrying out software engineering research (Montesi and Owen 2008). Generally speaking, journal publications or conference papers get citations, particularly high-quality journals and accredited conferences with a DOI and a credible publisher. Since both types of research articles are continued with researchers for disseminating knowledge but lead to different quality inferences. This paper, therefore, explores the potential impact differences between them. The hypothesis regarding the type of article is as follows.

**H2.** *Citation counts increase with the journal type of publication.*

### 2.3. First-Author Gender

Previous research on gender differences in academics confirmed the existence of gender differences in all region's studies and almost all disciplines; however, there are reports that such differences have decreased over time (Lynn et al. 2019). Such studies highlight several concerns, including research productivity (Hengel and Moon 2020), citations (Maddi and Gingras 2021), funding (Lerchenmueller and Sorenson 2018), and success in tenure (Mathews and Andersen 2001; Corley and Gaughan 2005; Weisshaar 2017); they found that women scholars are less productive than their male peers regarding publication numbers (Kyvik 1995; Cole and Zuckerman 1984; Lee and Bozeman 2005; Peñas and Peter 2006). Some research proves that women's publications are cited on average less than those of men (Turner and Mairesse 2005; Aksnes et al. 2011), even when evaluating articles or abstracts, concluding that males do better (Knobloch-Westerwick et al. 2013;

Krawczyk and Smyk 2016). Abramo et al. (2009) confirmed significant gender differences in research production using bibliographic indicators on male and female academics working at Italian universities in science and technology. Nevertheless, the differences were smaller than those reported in previous studies, and they also indicated that the effects of gender differences decreased over time. The topic of gender disparities in publication authorship and citation impact was examined, concluding that female first authors are cited more diminutive than their male counterparts (Larivière et al. 2013), which is in line with the study indicating that publications with male first authors received 3.6% more citations in India (Thelwall 2018). While research indicates that women's academic performance is of inferior quality to men, some research argues that there is no difference between male and female authors regarding the impact of publications (Long and Fox 1995; Bordons et al. 2003; Mauleón and Bordons 2006; Gonzalez-Brambila and Veloso 2007). Other studies pointed out that females' research productivity is better than that of males in science (Long 1992; Borrego et al. 2010) and uncovered that the effect of women's patents is greater than that of men (Whittington and Smith-Doerr 2005). Tower et al. (2007) studied the top six international journals in science, business, and social sciences and concluded that the gender difference in academic efficacy was not found when considering the percentage of women participating in the educational institution. The participation rate of women is 30–35% in academic positions, and women make up nearly 30% of the authors in the top journals; they also found that the gender gap does not exist considering the Journal Impact Factor based on their results. Therefore, the difference in quality is not due to gender differences; it is more of a disciplinary issue. In this context, the hypothesis regarding first author gender is as follows:

**H3.** *Citation counts increase with the first author being female.*

### 2.4. Journal's Scientific Prestige: The SJR Indicator

In understanding the nature of citations, Merton's normative citation theory states that these documents are cited when they influence the reader (Merton 1973). The references in the article to the importance and usefulness of the results presented should be directly correlated in principle. Subsequently, several factors other than cognitive influence and peer perception were identified, such as reputation and respect; it is a factor that increases not only the likelihood of academic promotion (Petersen et al. 2014) or research funding (Bol et al. 2018), but it can also influence readers of the work (Petersen et al. 2014). In other words, the more successful the researchers at the initial stage, the higher the chances of getting a publication citation, which to help them achieve even more success later on; this phenomenon is known as the Matthew Effect (Merton 1968). The SJR is a size-independent reputation and respect proxy that ranks journals by their average prestige per article, based on the perception that all citations are not created equal. Publishing in a high SJR score journal can be defined as the early success of the researcher. With SJR, the journal's subject field, quality, and prestige influence the citations (Bollen et al. 2006). The SJR, short for SCImago journal rank, is an index offered by Scopus as an alternative to measuring journal quality in addition to Thomson Reuters' Impact Factor (Falagas et al. 2008; Leydesdorff 2009); it was developed in 2009 by Prof. Félix de Moya in collaboration with researchers at SCImago Research Group, Spain, and is a metric that ranks journals based on journal articles and citations from Elsevier's Scopus database (Guz and Rushchitsky 2009); it applies the principle of the PageRank algorithm, which is the same method Google uses to rank and measure the importance of popular web pages (Page et al. 1999). The journal's weight in which the article is cited normalizes the journal citation value. For example, a journal in science with many citations has a lower weight than a social science journal with fewer citations. Therefore, journal articles referred by journals with higher SJR values will also receive higher SJR values. The SJR is calculated as the total number of citations a journal has received in the current year divided by the number of articles published in the past three years, with only selected articles and citations received from peer-reviewed papers, including research papers, articles review, and articles from the conference report. The

SJR will also benefit smaller journals, journals in languages other than English, and newly released journals in the Scopus database apart from Thomson Scientific (Gasparyan 2011). The investigation of whether citations are affected by the early success is a noteworthy determination for authors wishing to boost their research impact. Therefore, the hypothesis under the scientific prestige is as follows:

**H4.** *Citation counts increase with the SJR value.*

### 2.5. Cross-Institutional Research Collaboration

Combining knowledge and skills among experts from different fields allows research teams to work on multiple projects simultaneously; it enables researchers to expand the scope of research topics to cover human issues. Research and development (R&D) is an integral part of the Thai government's support to lift Thailand from the middle-income trap, the economic crisis, and the increasing economic challenges. As the problems in society worldwide become more complex, such as emerging infectious diseases, global warming, and food shortages, using only knowledge from a particular field of study, from a particular agency, or a particular sector may not solve the problem (Hall et al. 2008). The creation of cooperation in research benefits researchers and institutions in terms of strength of work, knowledge exchange, planning, and mutual support, resulting in the continuation of the body of knowledge and the integration of resource utilization for cost-effectiveness (Sonnenwald 2007), and the increase in research productivity (Beaver 2001; Thorsteinsdóttir 2000). In addition, knowledge exchange and extension of the research work create issues for continued development (Cvitanovic 2015). Recently research productivity yielded by collaboration includes solutions to water contaminated with lead in Flint (Lewis and Sadler 2021); conservation strategies for heritage sites by future climatic uncertainty (Richards et al. 2020); a community-academic partnership in insects' identification in a dense fragment of lowland rainforest (Paliau et al. 2022); and the R&D of vaccine funding by industry-academic collaboration (Cross et al. 2021).

The level of cooperation in research can be divided according to the working context, i.e., organizational, local, national, and international cooperation, in an integrated manner between experience and expertise (Song et al. 2019), and may collaborate methods and budgets. What needs to be concerned is the coordination and linkage of work between institutions to create a system that facilitates the promotion and expansion of scientific research production so that research can be utilized for economic and social development. Universities as educational institutions and learning centers for the development of knowledge and innovation are fundamental goals that the government desires to support and encourage more research and innovation to meet the country's growth needs. In this case, the university's faculty and researchers are essential drivers in leading the university to produce products that meet the country's needs. Innovating and publishing research can also help support this aim to become a world-class university; therefore, researchers should have the opportunity to cooperate with researchers/academics from outside institutions who have different bodies of knowledge and perspectives from different fields and sectors to discover scientific knowledge or create innovations. Inter-institutional collaboration can fill in the areas the individual research team lacks, such as skills, knowledge, budget, and technology. In this case, expanding the collaborative research network also means increasing the budget and valuable resources. As a result, government agencies constantly promote cross-disciplinary, -organizational, and -sector collaboration to achieve effective academic productivity (Hall et al. 2012). Therefore, researchers should not limit themselves to the framework of their institutions but also seek externally as these will create a more outstanding value.

Researchers within teams come from different institutional affiliations and have a wide range of impacts, including exchanges, patents, expertise in specialized tools, software, and critical information from individuals. Literature suggests that the number of affiliated institutions positively impacts referrals (So et al. 2015); it raises the question of whether a greater diversity of institutions within the research team can positively impact research

productivity? Thus, the hypothesis regarding cross-institutional research collaboration is as follows:

**H5.** *Citation counts increase with the number of affiliated institutions for a publication.*

*2.6. International Research Collaboration*

Combining human capital and physical resources from different backgrounds and cultures benefits the production of scientific knowledge. International Research Collaboration (IRC) is considered a driving force to strengthen intellectual capital and boost innovation. Creating a research network in the form of a complete network integration leaves room for senior and junior researchers to collaborate as a team. The assistance and knowledge exchange in the network, both domestically and internationally, strengthen the research groups and stimulates them to deliver high-quality, rapid, concrete research results that benefit researchers to be developed in all aspects by leaps and bounds. Collaboration in research, particularly international cooperation, is widely recognized as beneficial to both researchers and related organizations as it improves research quality (Katz and Hicks 1997; Van Raan 1998; Leydesdorff and Wagner 2008; Van den Besselaar et al. 2012), resulting in more publications and citations (Glänzel and Schubert 2001; Glänzel 2001; Hara et al. 2003; Persson et al. 2004; Haustein et al. 2015; Wesel 2016; Fox et al. 2016). A positive effect of international collaboration has been found, either on research efficiency or average output quality, based on the international collaboration of researchers from Italian universities from 2001 to 2005, and each researcher is the analytical unit (Abramo et al. 2011). Academics from leading countries in academic circles such as the United States of America, the United Kingdom, and Australia attract researchers worldwide by collaborating with academics from these countries. At the same time, China and Germany are increasingly influential as leading research-producing countries in the leading general higher education journals (Fu et al. 2022). Therefore, the existing co-authorship model was strengthened. The most successful teams in terms of quantity and quality consist of members from different countries, and there is moderate team diversity since researchers leverage knowledge from different research cultures and their implications (Barjak and Robinson 2008).

In comparison between domestic and international papers, previous studies have concluded that internationally staffed publications are cited more often than domestically collaborative publications (Narin et al. 1991; Schmoch and Schubert 2008; Sooryamoorthy 2009). As international collaboration is considered to increase publication visibility and efficiency, the impact of having foreign researchers on research cooperation continues to be scrutinized. Academics in developing countries especially appreciate international cooperation because international collaborative papers attract readers' attention and are often cited in high-ranking journals rather than articles without cooperation internationally (Cronin and Shaw 1999; Moed 2005). The way to organize international cooperation has also impacted referrals; that is to say, it organizes which institutions or countries lead the collaboration and which groups play secondary roles. Because high and low citation countries cooperate, such research concludes that the order of authors is truly important. When authors from high-impact countries came before, 67% of international bilateral collaborative research had an average citation impact greater than the average citation of purely domestic papers from cooperating countries. On the other hand, the order of authors from low-impact countries proceeds first, leading to this percentage dropping to 43%. This concept is incorporated into the recently developed indicators—the so-called "research guarantor" by Moya-Anegón et al. (2013); however, other studies offer evidence that publications with international cooperation are not highly cited (Gazni and Didegah 2011; Rey-Rocha et al. 2001) but that the difficulty in managing International cooperation may be reduced by bringing researchers closer together geographically (Cummings and Kiesler 2005).

In order to measure the diversity indices for internationality, the Simpson's Diversity Index is used to measure the degree of concentration in a community when individuals are classified into types (Simpson 1949). The international diversity index for each research team was calculated as follows:

$$Simpson's\ Index = \frac{\sum n_i(n_i - 1)}{N(N-1)} \tag{1}$$

$$Simpson's\ Index\ of\ Diversity = 1 - \frac{\sum n(n-1)}{N(N-1)} \tag{2}$$

where:

$n_i$ = the total number of members that belong to species *i*;
$N$ = the total number of members.

Simpson's Diversity Index value is between 0 and 1, where 1 represents no diversity and 0 indicates maximum diversity. The higher the value, the lower the diversity. Foreign researchers within the research team have a higher impact in terms of publication that tends to attract more citations than domestic collaboration (Narin et al. 1991). Some argue that foreign researchers on the research team do not result in higher citations (Rey-Rocha et al. 2001). Thus, the hypothesis regarding International Collaboration is as follows:

**H6.** *Citation counts increase with the first author being a foreigner.*

**H7.** *Citation counts increase with the high international diversity of the research team.*

### 3. Methodology

*3.1. Data*

Chiang Mai University has divided the disciplines offered at the university to facilitate the university's administration and comply with the regulations referred to in the university's field of study; it covers all disciplines offered in faculties and colleges into three groups as follows: (1) Science and Technology Discipline Group, namely the Faculty of Science, the Faculty of Engineering, the Faculty of Agriculture, the Faculty of Agro-Industry, the Faculty of Architecture, the College of Marine Studies and Management, the College of Art, Media and Technology, and the International College of Digital Innovation; (2) Group of Health Sciences, namely the Faculty of Medicine, the Faculty of Dentistry, the Faculty of Pharmacy, the Faculty of Associated Medical Sciences, the Faculty of Nursing, the Faculty of Veterinary Medicine, and the Faculty of Public Health; (3) Group of humanities and social sciences, namely the Faculty of Humanities, the Faculty of Education, the Faculty of Fine Arts, the Faculty of Social Sciences, the Chiang Mai University Business School, the Faculty of Economics, the Faculty of Law, the Faculty of Mass Communication, and the Faculty of Political Science and Public Administration (Office of Educational Quality Development 2017).

Since the nature of each discipline is not the same, the nature of citations (e.g., citation characteristics and trends) in each discipline is different as a result (Batista et al. 2006). Some fields have a more expansive influence than others, so evaluating scientific research productivity is generally best viewed in context within a particular field of study. Therefore, journals in different disciplines should not be compared; they should be compared in the same group (Gates et al. 2019). In this case, the data used in the study were Chiang Mai University's science and technology research field published on the Scopus database from 2010 to 2019, with 3883 publications. The papers were collected from the Scopus database to cover the full range of researchers affiliated with Chiang Mai University. The detailed information of the publication was collected (i.e., article name, citation number, author number, abstract, keywords, publication type, publisher, volume, issue, start page, end page) and the authors' information (i.e., author name, affiliation, gender, and country of institution). The research team, a collaborative group of Chiang Mai university researchers, contributing to the publications, is represented as the main unit of analysis.

*3.2. Variables*

Research productivity is an essential indicator for evaluating effectiveness in research production. Research is like an input-output process, and results can be intangible (e.g., knowledge, skills, and consulting activities) and tangible (e.g., publications, patents). Measuring research efficiency is a complex task, and accurate measurements are important because researchers' past publications are considered in productivity assessment, impacting salaries and promotion. Universities and research institutes often use the number of publications and citations to measure the performance of researchers and research teams. The government considers the published results in allocating research funds at the macro-level (Gonzalez-Brambila and Veloso 2007). In general, research productivity measures can be measured in quantity and quality. The quantitative aspect can be measured by the publication rate, while the number of citations reflects quality research productivity.

However, there is some argument over the effectiveness of the citation index (Hirsch 2007), which addresses widely used alternative indices, such as journal impact factors and the H index. The Impact Factor is a tool to compare and rank journals. Typically, an article chosen for publication in a journal is considered a seal of approval by the academic community, certifying that an article has been reviewed by both the journal's editors and peer reviewers; thus, it can also reflect the quality of research published in a journal. H index indicates the number of publications per researcher or citations per publication. Although the H-index attempted to address the weaknesses of the citation index, it found some disadvantages, such as interoperability (Petersen and Succi 2013); it did not consider the context of the reference, how it is used, and whether or not it weighs. For instance, it is just a reference for information without confirming or proving anything. Therefore, citation analysis remains an important metric used for quality assessment (Schmoch and Schubert 2008). For example, citation-based metrics remain the basis for tenure decisions, funding, and measuring journals' relative status (Ioannidis 2008).

The frequency of citations from credible databases increasingly influences the scientific evaluation process, from individual publication ratings to author evaluations and journal ratings. Engaging prediction variables were the number of authors, type of article, the femininity of the first author, foreign nationality of the first author, SJR value, and the degree of diversity of the country to which the researchers are affiliated. Regarding the dependent variable, citation data was collected from the Scopus database of Chiang Mai University's researchers' science and technology research domain from 2010 to 2019, a total of 10 years. The cutoff date for the Scopus citation count is December 31, 2019. Regarding a reference group for dummy variables used in this study, a conference is defined as a reference group for publication type; a male researcher is defined as a reference group for gender; a Thai researcher is defined as a reference group for nationality.

*3.3. Model Specification*

In the analysis of inferential statistics, multiple linear regression analysis is used in the case of parametric statistics. Quantile Regression (QR) and Ordinary Least Squares Regression (OLS) are used in the case of non-parametric statistics. The specification of the research productivity equation is:

$$
\begin{aligned}
citations_i = \ &\beta_0 + \beta_1 teamsize_i + \beta_2 typejour_i + \beta_3 femalefirst_i \\
&+ \beta_4 interfirst_i + \beta_5 sjr_i + \beta_6 numaff_i + \beta_7 interdiver_i + e_i
\end{aligned}
\tag{3}
$$

where:

*citations* = the number of citations received by the publishing article;
*teamsize* = the number of author(s) per paper;
*typejour* = a journal type of publication, described with a value of 1;
*femalefirst* = female as a first author, described with a value of 1;
*interfirst* = being foreigners as a first author, described with a value of 1;
*sjr* = the journal's SJR indicator, which is a numeric value;

*numaff* = the number of affiliation(s) in the research team;

*interdiver* = the international diversity per paper; value ranges from 0 to 1.

## 4. Results and Discussion

### 4.1. Authorship Pattern

The descriptive analysis of authorization patterns in science and technology research during the study period demonstrated that the collaborative research was greater than single-author publications. Of total publications, only 88 of 3883, or 2.27%, were single-author publications, while 3795 papers were co-published.

According to Table 1, team sizes range from one to more than 20 authors; two author publications accounted for 11.43% of total papers, and three authors' articles accounted for 14.65% of publications. The maximum number of publications is four authors with 15.86%. From the four written down, the percentage of the number of publications has continued to drop noticeably. The 21 authors above have published a total of 48 publications (1.24%) during ten years of study; it revealed that collaborative research dominates CMU's science and technology research.

**Table 1.** Chiang Mai university's authorship pattern of publications trends in science and technology.

| No. | No. of Author | Total No. of Publications | Percentage of 3883 |
|-----|---------------|---------------------------|--------------------|
| 1 | 1 author | 88 | 2.27 |
| 2 | 2 authors | 444 | 11.43 |
| 3 | 3 authors | 569 | 14.65 |
| 4 | 4 authors | 616 | 15.86 |
| 5 | 5 authors | 554 | 14.27 |
| 6 | 6 authors | 458 | 11.80 |
| 7 | 7 authors | 315 | 8.11 |
| 8 | 8 authors | 237 | 6.10 |
| 9 | 9 authors | 155 | 3.99 |
| 10 | 10 authors | 122 | 3.14 |
| 11 | 11 authors | 71 | 1.83 |
| 12 | 12 authors | 64 | 1.65 |
| 13 | 13 authors | 42 | 1.08 |
| 14 | 14 authors | 27 | 0.70 |
| 15 | 15 authors | 18 | 0.46 |
| 16 | 16 authors | 20 | 0.52 |
| 17 | 17 authors | 12 | 0.31 |
| 18 | 18 authors | 9 | 0.23 |
| 19 | 19 authors | 7 | 0.18 |
| 20 | 20 authors | 7 | 0.18 |
| 21 | >20 authors | 48 | 1.24 |
| Total | | 3883 | 100 |

### 4.2. Co-Authorship Pattern

In examining the authorship patterns of science and technology research, Table 2 shows the co-authorship pattern of Chiang Mai University's research team in the science and technology domain from 2010 to 2019. The author pattern is divided into five possible co-authors groups, e.g., a single author or a collaboration of two, three, four, and more than five authors. The total number of publications is 3883, with a total number of authors of 23,927 based on our data. Regarding the year of publication, the number of authors involved in science and technology, especially research teams of more than five co-authors, tends to increase steadily. So far in 2019, the number of publications has declined across all groups, decreasing the total publication number in that year. In the case of a single author, the results showed significantly lower numbers than in co-authored research, where the joint participation of more than five authors was remarkably high. Other metrics, such as the degree of collaboration (DC), support the fact that typical practice collaboration between

authors and collaborative research has rapidly increased in Chiang Mai University's science and technology research. The collaboration level of authorship is calculated using the following formula (Subramanyam 1983):

$$DC = \frac{NM}{NM + MS} \tag{4}$$

where:

$DC$ = Degree of Collaboration;
$NM$ = Number of Multi-authored articles;
$MS$ = Number of Single author articles.

**Table 2.** Chiang Mai university's co-authorship pattern from the year 2010 to 2019.

| Year of Publication | Single Author | Two Authors | Three Authors | Four Authors | ≥Five Authors | Total Publication | Total Authors | DC |
|---|---|---|---|---|---|---|---|---|
| 2010 | 3 | 41 | 38 | 47 | 97 | 226 | 1089 | 0.987 |
| 2011 | 8 | 36 | 50 | 53 | 146 | 293 | 1589 | 0.973 |
| 2012 | 8 | 33 | 63 | 65 | 179 | 348 | 1831 | 0.977 |
| 2013 | 7 | 36 | 57 | 62 | 189 | 351 | 1880 | 0.980 |
| 2014 | 14 | 51 | 64 | 58 | 185 | 372 | 1906 | 0.962 |
| 2015 | 7 | 62 | 69 | 58 | 235 | 431 | 2624 | 0.984 |
| 2016 | 11 | 56 | 65 | 74 | 266 | 472 | 3254 | 0.977 |
| 2017 | 8 | 51 | 58 | 87 | 297 | 501 | 3543 | 0.984 |
| 2018 | 17 | 57 | 73 | 77 | 369 | 593 | 3959 | 0.971 |
| 2019 | 5 | 21 | 32 | 35 | 203 | 296 | 2252 | 0.983 |
| Total | 88 | 444 | 569 | 616 | 2166 | 3883 | 23927 | 0.978 |

*4.3. International Research Collaboration*

To better understand the trend of international cooperation in science and technology research of researchers at Chiang Mai University, Table 3 depicts the co-authoring patterns of different countries. The data are sorted by the largest number of authors in each country from the total number of publications and was classified as paper with two authors, multiple authors, and Mega authors. Comparative results by country show that the United States, Japan, China, Australia, United Kingdom, Germany, France, India, South Korea, and Italy were among the top ranks in collaboration with CMU researchers. About 60% of all papers in the dataset come from the presented countries through the collaboration of two and more than two authors. The cross-country collaboration is depicted in Figure 1. The collaboration network shows that most countries collaborated partner is with countries such as the United States, Japan, and China. Figure 1 shows the results of international collaboration by continent that cooperates with CMU researchers in science and technology; it exhibits that Asian countries have the most cooperation with CMU researchers accounting for 39%, European countries accounting for 32%, and North America, 16%, respectively.

**Table 3.** Chiang Mai university's international authorship pattern in science and Technology.

| Country | Two Authors Articles | Three Authors Articles | Four to Ten Authors Articles | More Than Ten Authors Articles | Total |
|---|---|---|---|---|---|
| United States | 221 | 49 | 32 | 3 | 305 |
| Japan | 155 | 64 | 33 | 0 | 252 |
| China | 67 | 54 | 50 | 4 | 175 |
| Australia | 96 | 29 | 22 | 1 | 148 |
| United Kingdom | 103 | 19 | 7 | 0 | 129 |

**Table 3.** *Cont.*

| Country | Two Authors Articles | Three Authors Articles | Four to Ten Authors Articles | More Than Ten Authors Articles | Total |
|---|---|---|---|---|---|
| Germany | 78 | 16 | 12 | 0 | 106 |
| France | 45 | 8 | 12 | 0 | 65 |
| India | 31 | 22 | 8 | 1 | 62 |
| South Korea | 45 | 14 | 3 | 0 | 62 |
| Italy | 40 | 6 | 16 | 0 | 62 |
| Taiwan | 24 | 21 | 15 | 1 | 61 |
| Viet Nam | 39 | 7 | 3 | 0 | 49 |
| New Zealand | 43 | 3 | 1 | 0 | 47 |
| Iran | 19 | 6 | 19 | 0 | 44 |
| Other 71 Countries | 431 | 182 | 80 | 1 | 694 |
| Total | 1437 | 500 | 313 | 11 | 2261 |

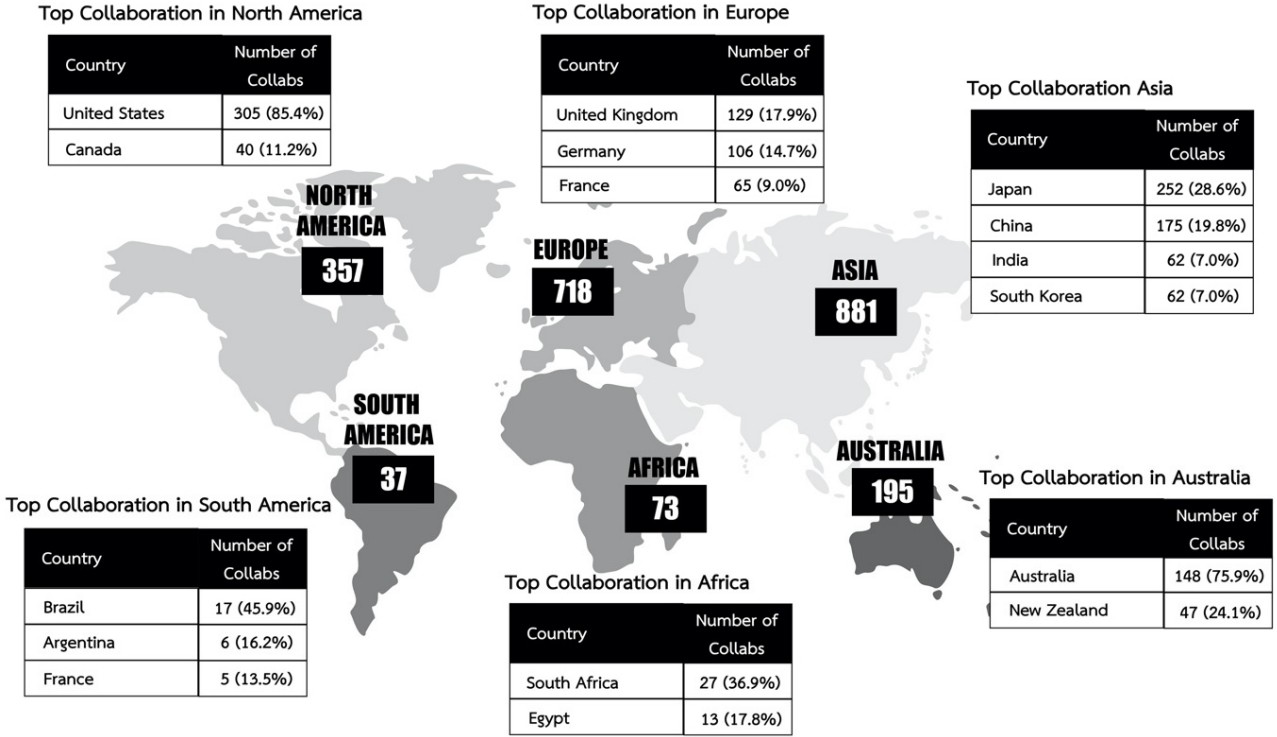

**Figure 1.** Chiang Mai university's cross-country collaboration in science and Technology by continent.

## 4.4. Authorship Pattern Based on Domestic Institutions

Cooperation pattern of domestic institutions that have networks with the CMU, as shown in Tables 4 and 5. From Tables 4 and 5, 15 institutions and authors' styles with the most contributions in science and technology research are; Prince of Songkla University, Mae Fah Luang University, Maejo University, University of Phayao, Suranaree University of Technology, Rajamangala University of Technology, Kasetsart University, Chulalongkorn University, Khon Kaen University, Naresuan University, Mahidol University, Rajamangala University of Technology Lanna, Ratchathani University, Ubon Ratchathani University, Srinakharinwirot University, respectively.

**Table 4.** Top 16 Most Collaborative Institutions.

| No. | Collaborative Institution | No. of Collaborated (Percentage) |
|---|---|---|
| 1 | Chiang Mai University | 1616 (45.16) |
| 2 | Prince of Songkla University | 197 (5.51) |
| 3 | Mae Fah Luang University | 193 (5.39) |
| 4 | Maejo University | 138 (3.86) |
| 5 | University of Phayao | 138 (3.86) |
| 6 | Suranaree University of Technology | 135 (3.77) |
| 7 | Rajamangala University of Technology | 114 (3.19) |
| 8 | Kasetsart University | 112 (3.13) |
| 9 | Chulalongkorn University | 111 (3.10) |
| 10 | Khon Kaen University | 89 (2.49) |
| 11 | Naresuan University | 68 (1.90) |
| 12 | Mahidol University | 67 (1.87) |
| 13 | Rajamangala University of Technology Lanna | 52 (1.45) |
| 14 | Ratchathani University | 38 (1.06) |
| 15 | Ubon Ratchathani University | 37 (1.03) |
| 16 | Srinakharinwirot University | 35 (0.98) |

**Table 5.** Chiang Mai university's domestic collaboration pattern by institutions.

| Institution | Two Authors Articles | Three Authors Articles | Four to Ten Authors Articles | More Than Ten Authors Articles | Total |
|---|---|---|---|---|---|
| Prince of Songkla University | 178 | 14 | 5 | 0 | 197 |
| Mae Fah Luang University | 164 | 28 | 1 | 0 | 193 |
| Maejo University | 131 | 4 | 3 | 0 | 138 |
| University of Phayao | 133 | 5 | 0 | 0 | 138 |
| Suranaree University of Technology | 127 | 7 | 1 | 0 | 135 |
| Rajamangala University of Technology | 107 | 7 | 0 | 0 | 114 |
| Kasetsart University | 79 | 21 | 12 | 0 | 112 |
| Chulalongkorn University | 68 | 11 | 32 | 0 | 111 |
| Khon Kaen University | 68 | 16 | 5 | 0 | 89 |
| Naresuan University | 59 | 9 | 0 | 0 | 68 |
| Mahidol University | 51 | 11 | 5 | 0 | 67 |
| Rajamangala University of Technology Lanna | 52 | 0 | 0 | 0 | 52 |
| Ratchathani University | 37 | 1 | 0 | 0 | 38 |
| Other 57 Institutions | 487 | 19 | 4 | 0 | 510 |
| Total | 1741 | 153 | 68 | 0 | 1962 |

The collaborative pattern findings show that the percentage from within CMU collaboration papers (45.16%) is higher than in collaboration with other institutes. The top 15 domestic collaboration ranges from 35 to 197 papers. Prince of Songkla University is the most frequent collaborator with CMU. The pattern of two-author articles is the most frequent coordination (1741 papers or 88.75%) of domestic collaboration. Among the top 15 institutes, Rajamangala University of Technology Lanna is the institute that has never collaborated with CMU in the more than two collaborators pattern. The pattern of more than nine institutions collaborating with CMU did not exist.

*4.5. Findings*

4.5.1. The Least Squares Assumptions

The most common estimation method for linear models is the Ordinary Least Squares (OLS) method; however, the OLS estimation in the regression model is effective as a Best Linear Unbiased Estimator (BLUE). Therefore, to examine the characteristics of the data before analyzing the relationship, the variables used in the analysis were checked against five preliminary GM assumptions that would make the OLS estimate unbiased, as follows.

Linear in Parameters

The linearity condition can be verified by the graph representing the relationship of independent variables with dependent variables, shown in Figure 2; it was found that the red line was not a straight line along the graph's horizontal axis; therefore, it can be concluded that it is not a linear condition.

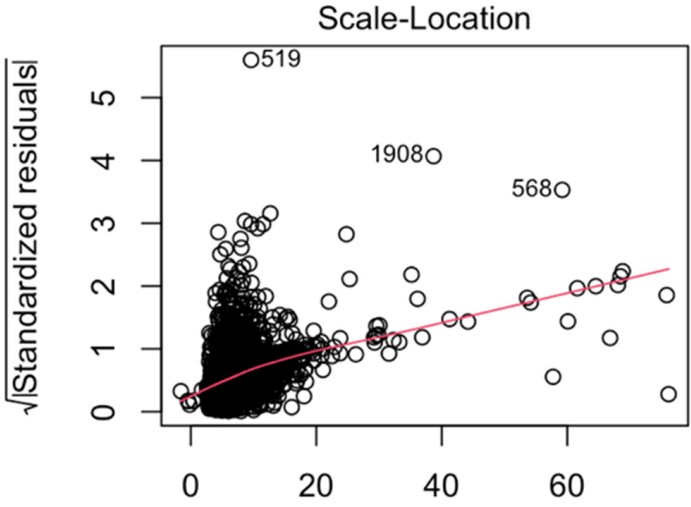

**Figure 2.** The linear relationship between independent variables and dependent variables.

Normal Distribution

The underlying random variable is normally distributed and is checked whether there is a normal distribution by using the histogram of citations, as shown in Figure 3 and statistical testing using the Kolmogorov–Smirnov Test. The histogram shows the number of citations and research at each journal level, with the vertical axis representing the number of data frequencies representing a non-normal distribution. From the Kolmogorov–Smirnov Test at the significance level of 0.05, the maximum deviation was 0.573, and the *p*-value was $2.2 \times 10^{-16}$, which is less than the specified level of significance. Therefore, it can be concluded that it is a non-normal distribution, which is in line with the graphical test.

Autocorrelation

Durbin–Watson statistics can test this preliminary agreement. The decision considering the statistical value from Durbin–Watson means there should be a value between 1.5–2.5 to be concluded that each error value is independent. In this case, considering the statistical value of Durbin–Watson, the value obtained from the test was found to be 1.7214, which is between 1.5–2.5. Therefore, it was concluded that individual error value is independent.

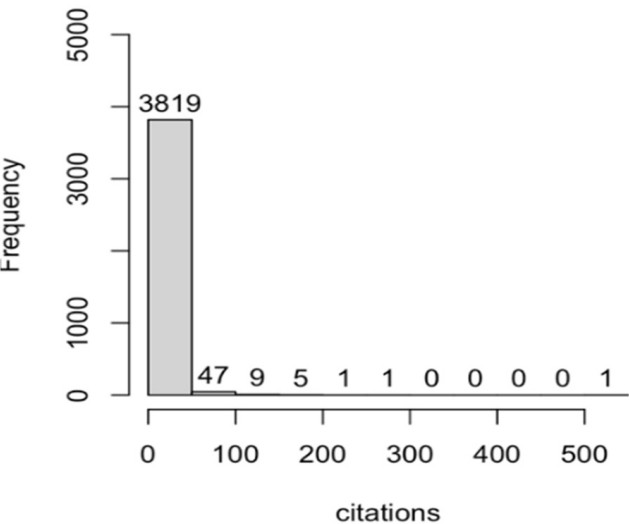

**Figure 3.** Histogram of citations.

Homoscedasticity

The error variance is assumed to be constantly tested by the residuals versus fitted value plot and the Breusch–Pegan Test method at a significance level of 0.05. According to Figure 4, the residuals versus leverage plot change shows that the data has a constant error variance; moreover, the test results using the method The Breusch–Pegan Test at a significance level of 0.05 showed a lower probability value than the defined significance level ($p$-value < 0.0001). Therefore, it can be concluded that the data has a constant error variance.

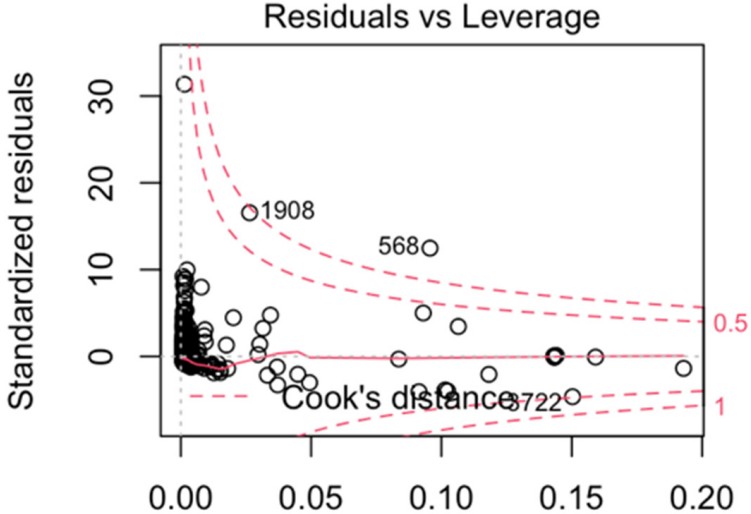

**Figure 4.** Residuals versus fitted value plot.

Multicollinearity

This assumption is validated using statistical values of Variance Inflation Factor (VIF) and Tolerance (TOL). There is a Multicollinearity problem if the Variance Inflation Factor is very close to 10. If the tolerance of a variable is close to 1, it means that the variables are independent of each other. The results of examining the relationship among independent variables are shown in Table 6.

**Table 6.** The statistical values of Variance Inflation Factor (VIF) and Tolerance (TOL).

| Variable | VIF | TOL |
|:---:|:---:|:---:|
| teamsize | 5.7094 | 0.1752 |
| typejour | 1.0027 | 0.9973 |
| femalefirst | 1.0033 | 0.9967 |
| interfirst | 1.1407 | 0.8766 |
| sjr | 1.0394 | 0.9621 |
| numaff | 5.9349 | 0.1685 |
| interdiver | 1.0021 | 0.9979 |

Table 6 presented that the Variance Inflation Factor of some variables is greater than 5, but the Tolerance of all variables is less than 1; it can be concluded that a relationship between the independent variables or a Multicollinearity problem is found. The tests used in the analysis to determine whether it was consistent with all five of the preliminary agreements revealed that it was non-compliant with the use of parametric statistics. This research, therefore, uses non-parametric statistics, including determination of the Spearman correlation coefficient and quantile regression analysis to suit such data.

4.5.2. The Impact of Authorship Characteristics on Citations

The outlier-tolerant quantitative regression is used as the primary regression to study the impact of a team of authors' characteristics on research productivity. To verify the robustness, a comparison of results from OLS regression and quantile regression for the median (0.5th quantile), the simplest quantile regression model that can be compared, is demonstrated. Since both methods attempt to simulate the central location of the response distribution, the interpretation of the regression coefficient of both methods is comparable; moreover, the quantile 0.5 gives the conditional median of the variable as it defines the independent variable and is considered a non-parametric alternative to linear regression tailored to the conditional mean. Thus, comparing OLS with the median regression is a common practice. In this study, we used the bootstrapping method to estimate the standard error since the i.i.d. does not support a change in response. Although the bootstrap point estimation is similar to the asymptotic approach, it gives an oblique standard error than the asymptotic standard error method. That is, bootstrap reported lower accuracy levels of quantile estimates of 0.5 than asymptotic estimates (Hao and Naiman 2007).

Stepwise Regression

The selection of the explanatory variable using statistical values, a statistics-based selection, is the process of inserting the source variable into the regression equation to select the factors with the highest relation to the dependent variable; and the statistically significant variable is added to the regression equation first. The regression equation with the explanatory variable from the above method describes the variance in the dependent variable the most, together with other statistically significant factors in the equation, making the regression model the best-predicted equation. Stepwise selection alternating between forward and backward is one of the methods of statistics-based selection, which is the step-by-step iterative construction of a regression model. The process involves selecting independent variables in a final model by adding or removing potential factors that meet the criteria for entry or removal in sequence. Based on statistical significance after each repetition, the process continues adding or removing variables and ends when the influence of all remaining explanatory variables in the equation is statistically significant. From stepwise regression, a new equation is obtained:

$$Citations_i = \beta_0 + \beta_1 teamsize_i + \beta_3 femalefirst_i + \beta_4 interfirst_i + \beta_5 sjr_i + e_i \quad (5)$$

We start by combining all variables in the same regression. The number of co-authors appears to be a significant predictor of publication quality. For example, a larger team may

consist of people with more senior and more experienced authors possibly writing higher impact articles. Women's first authors were associated with moderate citation counts and high-impact SJR values on citation counts. We first combine the two in the same regression (columns 2–3) and then consider the effects separately (columns 4–5). Therefore, the effect of variables on team attributes affecting research productivity is presented in Table 7.

**Table 7.** The impact of authorship characteristics on citations considering OLS and quantile regression (0.5th quantile).

| Variable | Full Model | | Reduced Model | |
|---|---|---|---|---|
| | OLS | Q (0.5) | OLS | Q (0.5) |
| teamsize | 0.216 *** | 0.253 *** | 0.384 *** | 0.193 *** |
| | (0.070) | (0.082) | (0.030) | (0.057) |
| typejour | 6.024 | 2.407 *** | | |
| | (6.063) | (0.584) | | |
| femalefirst | −1.183 ** | −0.456 ** | −1.165 ** | −0.370 ** |
| | (0.515) | (0.195) | (0.514) | (0.194) |
| interfirst | −0.493 | 0.086 | −0.164 | 0.212 |
| | (0.580) | (0.301) | (0.566) | (0.248) |
| sjr | 3.364 *** | 1.720 *** | 3.383 *** | 1.733 *** |
| | (0.312) | (0.367) | (0.312) | (0.374) |
| numaffiliation | 0.467 *** | −0.141 | | |
| | (1.174) | (0.128) | | |
| Interdiver | 0.809 | 0.163 | | |
| | (0.722) | (0.264) | | |
| Constant | −3.469 | −1.646 ** | 3.154 *** | 0.646 ** |
| | (6.056) | (0.652) | (0.459) | (0.285) |
| Observations | 3883 | 3883 | 3883 | 3883 |
| R Square | 0.079 | | 0.076 | |
| Adjusted R Square | 0.077 | | 0.075 | |
| Residual Std. Error | 16.004 | | 16.004 | |
| F Statistic | 47.197 *** | | 80.238 *** | |

Note: Standard errors are in parentheses; ** $p < 0.05$, *** $p < 0.01$.

Although authorship characteristics typically stay influential for citation counts, the findings were not always as hypothesized.

**H1** hypothesizes a positive relationship between the number of authors per paper and citations. The analysis results revealed a significant positive impact of the team size on citations ($p < 0.01$) at the 0.5th quantile and OLS model. The reduced model also confirms the strong relationship. The team size's coefficient in the quantile regression model, 0.193, is higher than that of the OLS's coefficient, which is 0.384. It could be implied that the new knowledge circulates faster with more authors and is expected to be better communicated to the relevant scientific community.

**H2** predicts a relationship between the journal type of publication and team research productivity. In column 3, the results showed a positive relationship ($p < 0.01$) between being journal papers and citations in the 0.5th-quantile regression model, which is 2.407; however, the impact does not exist in the OLS model. Therefore, the number of citations is higher for the journal type of publication than for the conference proceeding paper.

**H3** expects a positive correlation between female first-author papers and citations. In columns 2, 3, 4, and 5, the results indicated a significant negative impact of the teams with the female-first author on citations ($p < 0.05$) at the 0.5th quantile and OLS model; and is in line with the reduced model; it implies that the number of citations is lower for a female first author paper.

**H4** hypothesizes a positive impact of the SJR value on team research productivity. Considering columns 2, 3, 4, and 5, the analysis results using citations as the research productivity proxy indicated a positive correlation, both with OLS and QR (0.5); and is also

confirmed in the reduced model. Therefore, the number of citations increases when the SJR value increases by one, holding all the other factors constant.

**H5** expects that citation counts increase with the number of affiliated institutions for a publication. The results showed that the number of affiliated institutions positively impacts citations ($p < 0.01$) in the OLS regression model; however, such an impact does not exist in the median (0.5th quantile) model. In this case, data are insufficient to conclude that a paper with more institutional collaborators gains more citations.

**H6** predicted a negative relationship between the international first author paper and citations. The results from the conditional-median and conditional-mean models revealed that they fall short of statistical significance in both models and are in line with the reduced models.

**H7** hypothesizes a positive relationship between international diversity and research productivity. The conditional-mean and conditional-median models indicated a positive impact of international diversity on citations, but they are not statistically significant.

### 4.5.3. Estimation of Individual Conditional Quantiles

We also investigated the inner quantiles of the response variable's distribution in response to the predictor variables besides the median. The citation counts estimated from quantile regression across all quantiles are demonstrated in Table 8. Based on the results, the increasing number of authors in the team boosts citations over the entire distribution, except for the 0.25th quantile, with the most potent effect observed at 0.5–0.75th quantiles and a weak effect at the 0.99th quantile. Specifically, the co-authors' number does not matter to the relatively unproductive teams; it has little effect on the most productive teams, while it strongly impacts moderately productive groups. Many studies supported that the number of researchers within the team affects increasing referrals. In other words, the larger the number of researchers, the more likely it is to be cited (Bornmann and Daniel 2010; Annalingam et al. 2014; Biscaro and Giupponi 2014); this effect may be explained by the fact that more comprehensive networks involve longer author lists. Such a more extensive network allows readers to see the publication better and may cause possible future references (Liskiewicz et al. 2021); however, the outcome of high referrals should not be an impetus to create self-contained groups of researchers where individuals do not actually participate in the research. Instead, it is best to include a co-author with additional expertise, since diversity is the cornerstone for comprehensive analysis, leading to multidimensional utilization.

**Table 8.** Estimating a quantile regression across the distribution of citations.

| Variable | Q (0.25) | Q (0.50) | Q (0.75) | Q (0.99) |
|---|---|---|---|---|
| teamsize | 0.012 | 0.193 *** | 0.392 *** | 3.122 * |
| | (0.016) | (0.057) | (0.105) | (1.759) |
| femalefirst | −0.010 | −0.370 * | −1.161 *** | −26.223 ** |
| | (0.022) | (0.194) | (0.420) | (10.334) |
| interfirst | 0.449 *** | 0.212 | −0.102 | 5.022 |
| | (0.149) | (0.248) | (0.518) | (12.601) |
| sjr | 0.648 *** | 1.733 *** | 5.831 *** | 28.395 |
| | (0.142) | (0.374) | (0.674) | (19.350) |
| Constant | −0.134 *** | 0.646 ** | 2.565 ** | 38.466 * |
| | (0.051) | (0.285) | (0.586) | (13.512) |
| Observations | 3883 | 3883 | 3883 | 3883 |

Note: Standard error are in parentheses; * $p < 0.1$, ** $p < 0.05$, *** $p < 0.01$.

Publication in which the first author is female at quantile levels of 0.50, 0.75, and 0.99 were statistically significant; however, all coefficients were negative; it indicates that the independent variable has a negative relationship with the dependent variable at the statistically significant level of 0.1, 0.05, and 0.01, respectively; it was concluded that the average citations of a female first author are less than the average citations of an article with

a male first author, consistent with literature indicating that male first authors received more citations than female first authors (Larivière et al. 2013; Thelwall 2018). The citation advantage of being a foreign-first author only existed at the 0.25th quantile; however, there was no evidence of foreign-first author citation advantages or disadvantages for the moderate and very productive publications. The SJR values likewise have a strong positive impact on the three lowest quantiles; more SJR values influence low and medium-quality publications. A possible explanation for this somewhat misallocation of talent; the low proportion of females entering into science compared to nonscience careers leads to gender disparities in research productivity (Breda and Ly 2012).

## 5. Conclusions

While the government and institutions are increasingly focusing on cooperation in the production of scientific research, competent authors are expected to be able to conduct high-quality research that increases article discoverability and interestingness, resulting in a better impact; this trend encourages independently formed teams to interact with other associations to improve their work by taking criticism and critique into account (Stvilia et al. 2010). In cooperation between institutions, researchers' discipline, research problem areas, sharing technology, and the interaction of learning and management within research teams are essential to encourage researchers from one institution to work effectively with their peers from other institutions. Therefore, interpersonal skills are something that many researchers should have and need to practice because research can not only exist in universities, but also need to build networks of external cooperation. In addition to networking, maintaining relationships between institutions is imperative and always supports the research production's value (Boekholt et al. 2009). Joint research is more complex than single writing, and effective team management is required. Therefore, it is essential to understand the characteristics and diversity of research team compositions. This study endeavors to clarify the factors that determine the productivity of joint research. The number of citations is a baseline indicator of the team's performance, obtained from the Scopus database in a ten-year window between 2010 and 2019. We analyzed various factors affecting citations of papers published in science and technology by researchers affiliated with Chiang Mai University, using OLS regression with robust standard and Quantile Regression. Our empirical parts generate the following findings.

- The relationship between journal type of publication and citations is positive. The number of citations is higher for the journal type of publication than for the conference proceeding paper.
- Since the first author contributed more to the paper, we examine the gender issue and cross-country collaboration through the first-order position in the research production. Concerning quantile regression, we found a negative correlation between female first author papers and citation rate, conditional on the 0.5th, 0.75th, 0.99th, that is, the number of citations is lower for the paper with the female-first author, but such effect did not happen in the low productive papers. In determining the impact of the foreign-first author on citations, we found a positive relationship only in the low productive papers.
- With SJR value, we found a positive relationship with low to medium quantile citation rate. The number of citations increases when the SJR value increases except for the top productive papers.
- The conditional-mean model found a significantly positive association between the number of affiliated institutions and citations ($p < 0.01$); however, such an effect does not exist in the conditional-median model. Thus, data are insufficient to conclude that a paper with more institutional collaborators gains more citations.
- Finally, no relationship between international diversity and citations was found, since the analysis does not show any significant correlation. Although literature points out a positive effect of team size, institutions, and countries' collaboration on citation rates,

our data support the findings relevant to past research only the number of co-authors that influences citations.

This study sheds light on how authorship patterns and research team characteristics correlate to article-level citations, focusing on domestic and international research collaborations; however, our research also has some limitations. This study was only a research study in the Science and Technology group of Chiang Mai University, and the data were collected in 10 years time-bound. Some factors were omitted from the analysis (e.g., academic experience, gender diversity, and seniority diversity); they should be included in the model to provide a comprehensive view; moreover, literature proved that publishing in Open Access is beneficial to the citation frequencies compiled by a given publication (Liskiewicz et al. 2021).

Future studies should expand the disciplinary scope by investigating the productivity impact of other domains such as health sciences, humanities, and social sciences. The interval between the year of publication and the cutoff date for citation counting should be extended to allow enough time for some publications that are not cited. The investigation of factors beyond the scope of this research must be addressed, such as open access, the team's average H-Index, and differences in the institute's affiliation. According to the research team formation, an analysis of possible differences in the origin of the international authors is interesting to discover; such a social bias may exist when corresponding authors are from different countries (e.g., assumably biased toward English-speaking ones and vice versa). The factors mentioned earlier become more critical with alternative approaches to measuring the increased impact: future research may include reconstructive metrics (e.g., Eigenfactor, Impact Factor), social recognition, and therefore need a holistic approach. In addition, researchers may examine the impact of team composition in greater depth (e.g., the pattern and type of team relationships) by interviewing team members to observe and collect qualitative data (e.g., motivation for participating in the research team). The abovementioned recommendations enable further studies in accurate and comprehensive modeling of research productivity improvement topics to better understand the relationship between team composition and research efficiency.

**Author Contributions:** Conceptualization, B.P. and R.W.; methodology, B.P.; software, B.P.; validation, B.P.; formal analysis, B.P.; investigation, B.P.; resources, B.P.; data curation, B.P.; writing—original draft preparation, B.P.; writing—review and editing, B.P.; visualization, B.P.; supervision, R.W.; project administration B.P.; funding acquisition, B.P. All authors have read and agreed to the published version of the manuscript.

**Funding:** This research was supported by CMU Junior Research Fellowship Program.

**Institutional Review Board Statement:** Not applicable.

**Informed Consent Statement:** Not applicable.

**Data Availability Statement:** Not applicable.

**Acknowledgments:** The first author would like to express special thanks to Chayanut Phantharot for administrative and technical support.

**Conflicts of Interest:** The authors declare no conflict of interest.

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
