# Peer review of "Investigating Patterns of Research Collaboration and Citations in Science and Technology: A Case of Chiang Mai University"

_admsci, doi:10.3390/admsci12020071_

Round 1

Reviewer 1 Report

Dear authors,

I would like to congratulate you for such an extense work. 

The analysis carried out is excellent and it provides a clear idea of fhe publishing framework of your university. I like the approach you made by posing all the hypotheses with a detailed introduction,

Regarding the hypotheses I was surprised by the choice of hypothese H3 because my choice would be the opposite as you found out after developing your analysis. I have also a comment about the following sentence in this same paragraph: "No statistically significant differences were found in the JIF rankings for both sexes" I don't understand what you refer to

I also like the proposal for future work and I encourage you to carry out it. Moreover there is another analysis that could be relevant to perform: to analyse possible differences on the origin of the international authors because I suspect they exist when corresponding authors are from different countries probably biased to English speaking ones

Thanks for your work

Author Response

Dear reviewers:

We want to take this opportunity to express our sincere thanks to you and the reviewers for your time and effort in reviewing our manuscript. The feedback has been invaluable in improving the content and presentation of the paper. We were pleased to revise and re-submit the modified manuscript addressing all reviewers' concerns to Administrative Sciences.

I attached the revised version of the paper manuscript, plus our point-by-point response to the comments raised by the reviewers as outlined below in this letter. We agree with all the reviewers' comments and thus have carried out all revisions/corrections accordingly. All in-text alterations were marked up using the “Track Changes” function. All authors have read and approved the revised manuscript. We hope that our resubmission is now suitable for inclusion in Administrative Sciences, and we look forward to hearing from you.

Sincerely,

The authors

RESPONSES TO THE REVIEWERS' COMMENTS

All authors do agree with the comments raised by the reviewers and thus have carried out all revisions/corrections accordingly.

REVIEWER 1

Comment #1:

I have also a comment about the following sentence in this same paragraph: "No statistically significant differences were found in the JIF rankings for both sexes" I don't understand what you refer to.

Response #1:

Thank you very much for pointing this out. We agree with the comment that the sentence is not clear. Therefore, we have revised the sentence (lines 376-383) as:

“Tower et al. (2007) studied the top six international journals in science, business, and social sciences and concluded that the gender difference in academic efficacy was not found when considering the percentage of women participating in the educational institution. The participation rate of women is 30-35% in academic positions, and women make up nearly 30% of the authors in the top journals. They also found that the gender gap does not exist considering the Journal Impact Factor based on their results. Therefore, the difference in quality is not due to gender differences; it is more of a disciplinary issue.”

Comment #2:

I also like the proposal for future work and I encourage you to carry out it. Moreover there is another analysis that could be relevant to perform: to analyse possible differences on the origin of the international authors because I suspect they exist when corresponding authors are from different countries probably biased to English speaking ones.

Response #2:

Thank you for the valuable suggestion. It is interesting to explore this aspect. We have added the suggested content to the manuscript (lines 931-935) as:

“According to the research team formation, an analysis of possible differences in the origin of the international authors is interesting to discover. Such a social bias may exist when corresponding authors are from different countries (e.g., assumably biased toward English-speaking ones and vice versa).”.

Reviewer 2 Report

It seems to me a very extensive study, as evidenced by the nearly 4,000 investigated articles affiliated with Chiang Mai University.

Even though the authors say that they conducted the study for a period of 2010-2019, I still saw some more recent bibliographical references, including from the current year 2022.

I have only a few recommendations to improve the final look of the article for publication.

The manuscript seems to be an original work, proved by the precent of similarity in anti-plagiarism software (15%). There are some paragraphs that may need a refrain (they can be seen colored in the PDF file attached to this review).

I recommend that tables 1, 4 etc. not to be split on two separate pages, for a better understanding.

I recommend at least a small increase in the size of the figures 2,3,4 for a better view of the readers.

I recommend that section / sub-section titles not be positioned alone, right at the bottom of the page (as in the PDF file received for review): 4.3 International Research Collaboration, 4.3.3 Considering other ...

I would like to read (in this article) more about the authors' plans to improve their present study, and even about their future plans for continuing their research.

Author Response

Dear reviewers:

We want to take this opportunity to express our sincere thanks to you and the reviewers for your time and effort in reviewing our manuscript. The feedback has been invaluable in improving the content and presentation of the paper. We were pleased to revise and re-submit the modified manuscript addressing all reviewers' concerns to Administrative Sciences.

I attached the revised version of the paper manuscript, plus our point-by-point response to the comments raised by the reviewers as outlined below in this letter. We agree with all the reviewers' comments and thus have carried out all revisions/corrections accordingly. All in-text alterations were marked up using the “Track Changes” function. All authors have read and approved the revised manuscript. We hope that our resubmission is now suitable for inclusion in Administrative Sciences, and we look forward to hearing from you.

Sincerely,

The authors

RESPONSES TO THE REVIEWERS' COMMENTS

All authors do agree with the comments raised by the reviewers and thus have carried out all revisions/corrections accordingly.

REVIEWER 2

Comment #1:

Even though the authors say that they conducted the study for a period of 2010-2019, I still saw some more recent bibliographical references, including from the current year 2022.

Response #1:

Thank you for the valuable comment. The references from the current year are used as evidence to support the idea that the study of collaboration characteristics is still an interesting topic in recent years. Moreover, the recent literature also pointed out the additional factors to be included in the model, which are valuable to our future study.

Comment #2:

The manuscript seems to be an original work, proved by the precent of similarity in anti-plagiarism software (15%). There are some paragraphs that may need a refrain (they can be seen colored in the PDF file attached to this review).

Response #2:

Thank you for pointing this out. Therefore, we have incorporated your suggestion throughout the manuscript according to the colored sentence in the PDF file attached.

Comment #3:

I recommend that tables 1, 4 etc. not to be split on two separate pages, for a better understanding.

Response #3:

We have accordingly modified tables 1, 4, and 7 to be on the same pages.

Comment #4:

I recommend at least a small increase in the size of the figures 2,3,4 for a better view of the readers.

Response #4:

Thank you for pointing this out. According to your suggestion, we have enlarged the figures 2, 3, and 4.

Comment #5:

I recommend that section / sub-section titles not be positioned alone, right at the bottom of the page (as in the PDF file received for review): 4.3 International Research Collaboration, 4.3.3 Considering other ...

Response #5:

We have modified the section and sub-section titles (i.e., sub-section 4.3 and 4.5.3) to be positioned on the same page as its content throughout the manuscript.

Comment #6:

I would like to read (in this article) more about the authors' plans to improve their present study, and even about their future plans for continuing their research.

Response #6:

Thank you for the valuable suggestion. In this case, the future work’s part has been revised to emphasize this point. The revised content (lines 926-943) is:

 “Future studies should expand the disciplinary scope by investigating the productivity impact of other domains such as health sciences, humanities, and social sciences. The interval between the year of publication and the cutoff date for citation counting should be extended to allow enough time for some publications that are not cited. The investigation of factors beyond the scope of this research must be addressed, such as open access, the team's average H-Index, and differences in the institute's affiliation. According to the re-search team formation, an analysis of possible differences in the origin of the international authors is interesting to discover. Such a social bias may exist when corresponding authors are from different countries (e.g., assumably biased toward English-speaking ones and vice versa). The factors mentioned earlier become more critical with alternative approaches to measuring the increased impact: future research may include reconstructive metrics (e.g., Eigenfactor, Impact Factor), social recognition, and therefore need a holistic approach. In addition, researchers may examine the impact of team composition in greater depth (e.g., the pattern and type of team relationships) by interviewing team members to observe and collect qualitative data (e.g., motivation for participating in the research team).”.